# Anti-Inflammatory Effect of Ethanol Extract from *Hibiscus cannabinus* L. Flower in Diesel Particulate Matter-Stimulated HaCaT Cells

**DOI:** 10.3390/nu16223805

**Published:** 2024-11-06

**Authors:** Ji-Ye Han, Shin-Kyeom Kim, Do-Won Lim, Osoung Kwon, Yu-Rim Choi, Chan-Ho Kang, Yun Jung Lee, Young-Mi Lee

**Affiliations:** 1Department of Oriental Pharmacy, College of Pharmacy, Wonkwang University, Iksan 54538, Republic of Korea; 2Wonkwang-Oriental Medicines Research Institute, Wonkwang University, Iksan 54538, Republic of Korea; 3Division of Crops & Food, Jeonbuk-do Agricultural Research & Extension Services, Iksan 54591, Republic of Korea

**Keywords:** *Hibiscus cannabinus* L., kenaf, diesel particulate matter, atopic dermatitis, inflammation

## Abstract

Background/Objectives: Diesel Particulate Matter (DPM) is a very small particulate matter originating from cities, factories, and the use of fossil fuels in diesel vehicles. When DPM permeates the skin, it causes inflammation, leading to severe atopic dermatitis. *Hibiscus cannabinus* L. (Kenaf) seeds and leaves possess various beneficial properties, including anti-coagulation, antioxidant, and anti-inflammation effects. In this study, we investigated the anti-inflammatory effects of an ethanol extract of *Hibiscus cannabinus* L. flower (HCFE) in HaCaT cells stimulated with 100 μg/mL of DPM. Methods: The anthocyanin content of HCFE was analyzed, and its antioxidant capacity was investigated using the DPPH assay. After inducing inflammation with 100 ug/mL of DPM, the cytotoxicity of HCFE 25, 50, and 100 ug/mL was measured, and the inhibitory effect of HCFE on inflammatory mediators was evaluated. Results: Anthocyanin and myricetin-3-*O*-glucoside were present in HCFE and showed high antioxidant capacity. In addition, HCFE decreased the mRNA expression of inflammatory cytokines and chemokines such as IL-1β, IL-4, IL-6, IL-8, IL-13, and MCP-1, and significantly reduced the gene expression of CXCL10, CCL5, CCL17, and CCL22, which are known to increase in atopic dermatitis lesions. Furthermore, HCFE reduced intracellular reactive oxygen species (ROS) production, and down-regulated the activation of NF-κB, MAPKs. Inhibition of the NLRP-3 inflammasome was observed in DPM-stimulated HaCaT cells. In addition, the restoration of filaggrin and involucrin, skin barrier proteins destroyed by DPM exposure, was confirmed. Conclusions: These data suggest that HCFE could be used to prevent and improve skin inflammation and atopic dermatitis through the regulation of inflammatory mediators and the inhibition of skin water loss.

## 1. Introduction

Diesel particulate matter (DPM) is a very small particulate matter that results from the use of fossil fuels in cities, factories, and diesel vehicles. Rapid urbanization worldwide has led to an increase in the generation of DPM, which is currently the main cause of air pollution [1,2]. DPM poses a threat to human health, causing problems in the body’s major organs and systems, including the lungs, cardiovascular system, eyes, skin, and immune system [3,4]. In particular, DPM causes inflammatory diseases such as pneumonia and asthma and increases premature mortality [5,6]. Owing to the risk of DPM, the World Health Organization has presented guidelines for improving air pollution, and several countries have made efforts to reduce DPM and improve the atmospheric environment. Several studies have reported the health effects of DPM, which is known to trigger inflammation and oxidative stress [7].

Atopic dermatitis is a chronic inflammatory disease that occurs primarily in childhood and is characterized by extreme itching. It is common in children, with high recurrence rates, making it difficult to cure. Moreover, appropriate treatment is essential because it is the starting point of the atopic march that leads to asthma and allergic rhinitis [8]. Atopic dermatitis is a hypersensitive immune response and its pathogenesis is closely linked to inflammatory responses. Abnormalities in the type 2 inflammatory pathway caused by the activation of Th2 cell responses are a major clinical feature of atopic dermatitis [9]. IL-4 and IL-13 are the key factors that cause type 2 inflammation. If secreted continuously, it leads to itching through the activation of IL-31 and mast cells, causing the dysfunction of the skin barrier, and increasing skin infections caused by external factors [10]. Additionally, skin barrier disruptions, such as tight junction defects and reduced moisturizing factors, are commonly observed in atopic dermatitis lesions [11]. Tight junctions are complexes that connect cells and regulate the movement of ions and moisture between cells; filaggrin is a representative moisturizing factor that composes the epidermis [12]. In atopic dermatitis lesions, proteins that form tight junctions and the skin barrier are significantly reduced. Defects in these major skin components increase transepidermal water loss and cause skin dryness and itching, further worsening atopic dermatitis symptoms [13]. Topical steroids and biological agents that suppress type 2 inflammation are primarily used to treat atopic dermatitis. However, topical steroids and biological agents cause serious adverse effects such as skin atrophy, vasodilation, adrenal suppression, infectious disorders, and musculoskeletal disorders [14,15]. Therefore, better treatment with fewer adverse effects that can be safely used for a long period is needed.

Herbal medicine has been used traditionally for a long time and refers to utilizing the active ingredients derived from plant parts such as roots, leaves, and flowers. It is widely used by many people because it is easily accessible, has few side effects, and is inexpensive. In addition, as nature-friendly trends have grown, many products using plants have been developed, such as plant-based medicines, functional foods, and plant-based cosmetics [16]. Herbal medicines are attracting attention as substitutes for synthetic drugs because they are familiar to humans and have fewer adverse effects [17].

*Hibiscus cannabinus* L., also called kenaf, belongs to the family Malvaceae. It is actively cultivated in places such as Thailand, India, and Malaysia, and is an important fiber crop worldwide [18]. *H. cannabinus* has traditionally been used in rope, paper, and absorbent materials; however, recently, its availability as a functional food and pharmaceutical material has been increasing. According to previous studies, the seed, leaf, and flower extract of *H. cannabinus* contain various phenols, flavonoids, terpenes, and tocopherols; these demonstrate pharmacological activities, such as antibacterial, antioxidant, and anticancer properties [19,20,21]. However, the effect of *H. cannabinus* on skin diseases is still unknown.

In this study, we investigated the effects of *H. cannabinus* flower extract (HCFE) on human keratinocytes. Furthermore, we evaluated whether HCFE suppresses the expression of pro-inflammatory cytokines and regulates anti-inflammatory mechanisms in an in vitro model of DPM-induced skin inflammation.

## 2. Materials and Methods

### 2.1. Chemical and Reagents

First, 3-(4,5-Dimethylthiazol-2-yl)-2,5-Diphenyltetrazolium Bromide (MTT) was obtained from Duchefa Biochemie (Haarlem, The Netherlands), and 2,2-diphenyl-1-picrylhydrazyl (DPPH) was purchased from Sigma-Aldrich (St. Louis, MO, USA). The myricetin-3-*O*-glycoside was obtained from Extrasynthese (Genay, France). The cyanidin-3-*O*-glucoside was purchased from Chem Faces (Wuhan, China). Moreover, 5-(6)-Chloromethyl-2′,7′-Dichlorodihydrofluorescein diacetate (CM-H2DCFDA) was purchased from Thermo Fisher Scientific (Waltham, MA, USA). Also, Western blot primary antibodies were obtained from Cell Signaling Technology (Beverly, MA, USA) and the contents are as follows: phospho-ERK1/2 (#9101), ERK1/2 (#4695), phospho-JNK (#4668), JNK (#9252), phospho-p38 (#4511), p38 (#9212), phospho-JAK1 (#3331), JAK1 (#3344), phospho-STAT6 (#56554), STAT6 (#5397), phospho-p65 (#3033), p65 (#8242), phospho-IκBα (#2859), IκBα (#9242), α-tubulin (#2144), Lamin B1 (#13435), NLRP3 (#15101), IL-1β (#12703), and β-actin (#4970). Caspase-1 (sc-56036), filaggrin (sc-66192), and involucrin (sc-21748) were purchased from Santa Cruz Biotechnology (Santa Cruz, CA, USA). Filaggrin and NLRP3 antibodies were used at a dilution of 1:500, and all other antibodies were used at a dilution of 1:1000.

### 2.2. Preparation of Diesel Particulate Matter (DPM)

DPM was purchased from Sigma-Aldrich (#NIST2975, St. Louis, MO, USA). The mean diameter of DPM particle size was 31.9 ± 0.6 (volume distribution, MV, μm). The DPM solution was prepared by modifying the method of Hong et al. [22]. DPM was dissolved in 50% dimethylsulfoxide (DMSO) at room temperature for 24 h using a magnetic stirrer. Then, the solvent was filtered through a 40 μm filter and stored at 4 °C until use. DMSO was used at a final concentration of less than 0.1% (*v*/*v*).

### 2.3. Preparation of Hibiscus cannabinus L. Flower Ethanol Extract (HCFE)

*Hibiscus cannabinus* L. flowers were provided by Jeonbuk-do Agricultural Research & Extension Services (Iksan, Republic of Korea). Dried *Hibiscus cannabinus* L. flowers 30 g were added to 300 mL of 70% ethanol and soaked at room temperature for 72 h; the process was repeated twice. The solvent was filtered with a 6 μm filter paper (ADVANTEC, Tokyo, Japan) and concentrated in a rotary vacuum evaporator (EYELA N-1110, Tokyo, Japan) at 40 °C. Freeze-drying was then performed at −80 °C to obtain the HCFE powder, and the final yield of the obtained powder was 31.67%. The power was stored at −20 °C until use. HCFE was dissolved in DMSO for the treatment of cells. The final concentration of DMSO used on cells was less than 0.1%.

### 2.4. High-Performance Liquid Chromatography Analysis of Hibiscus cannabinus L. Flower Ethanol Extract (HCFE)

For sample analysis, the standard and HCFE were diluted and prepared. The myricetin-3-*O*-glycoside standard solutions were prepared in 70% ethanol at a concentration of 6.25, 12.5, 25, 50, and 100 μg/mL. The HCFE solution was prepared in 70% ethanol at a concentration of 1 mg/mL and filtered using a 0.2 μm polytetrafluoroethylene syringe filter (Ø13mm, Hyundai micro, Seoul, Republic of Korea). Ultra performance liquid chromatography (UPLC) was performed using an ACQUITY UPLC H-Class system with a PDA detector (Waters, Milford, MA, USA). The analytical column was an ACQUITY UPLC BEH C18 Column, 1.7 µm, 2.1 mm × 100 mm (Waters, Milford, MA, USA) with an ACQUITY UPLC BEH C18 VanGuard Pre-column, 1.7 µm, 2.1 mm × 5 mm (Waters, Milford, MA, USA). The mobile phases consisted of (A) 0.1% formic acid in water and (B) 0.1% formic acid in acetonitrile. The gradient program was set as follows: 0–4 min 14–18% B, 4–8 min 18–46% B, 8–10 min 46–100% B, 10–12 min 100% B, 12–14 min 100–14% B, 14–20 min 14% B. The mobile phase flow rate was 0.3 mL/min, and the injection volume was 1.5 μL. The column temperature was maintained at 30 °C, and the extract was detected by PDA at the wavelength of 356 nm.

### 2.5. Analysis of Total Anthocyanin Content

The total anthocyanin content of HCFE was determined using the pH differential method [23]. First, potassium chloride (0.03 M, pH 1.0) and sodium acetate (0.4 M, pH 4.5) were prepared. Then, 100 mg of HCFE was dissolved in 1 mL of distilled water. It was diluted 100-fold in each solution, to a final volume of 2 mL. The absorbance of the diluted HCFE was measured at 520 and 700 nm, and distilled water was used as the control for the diluted HCFE. The total anthocyanin content was expressed as mg cyanidin-3-*O*-glucoside equivalent per 100 mg HCFE (mg cy-3-glu/100 mg HCFE), and the calculation formula was as follows:$$\mathrm{Total}\;\mathrm{anthocyanin}\;\mathrm{content}\;\left(\mathrm{mg}/100\;\mathrm{mg}\right)=\frac{A^{\prime }\times \mathrm{MW}\times \mathrm{DF}\times 10^{3}}{\varepsilon \times 1}$$
where *A*’ is absorbance = (520 nm pH 1.0 − 700 nm pH 1.0) − (520 nm pH 4.5 − 700 nm pH 4.5), *MW* is the molecular weight of cyanidin-3-*O*-glucoside (449.2 g/mol), *DF* is the dilution factor, and *ε* is the extinction coefficient for cyanidin-3-*O*-glucoside = 26,900 (L∙cm^−1^∙mol^−1^). The absorbance was measured using a microplate reader (SpectraMax 190, Molecular Devices, San Jose, CA, USA).

### 2.6. DPPH Radical Scavenging Assay

The antioxidant activity of HCFE was evaluated using the DPPH radical scavenging assay. The experimental method was slightly modified from that described by Frezzini et al. [24]. The DPPH was dissolved in absolute ethanol to a final concentration of 100 μM and filtered using a 0.2 μm filter. Ascorbic acid was used to compare the antioxidant activities of HCFE. The 12.5, 25, 50, and 100 μg/mL of HCFE and 20 μg/mL of ascorbic acid were prepared by dissolving each in ethanol. Then, HCFE was mixed with DPPH and incubated at 37 °C for 30 min. The absorbance was determined at 517 nm using a microplate reader (SpectraMax 190, Molecular Devices, USA), and the percentage of antioxidant activity was calculated using the following formula:*DPPH radical scavenging* (%) = [(*Ac* − *As*) ÷ *Ac*] × 100
where *Ac* is the absorbance of the control and *As* is the absorbance of the sample.

### 2.7. Cell Culture

The human keratinocyte (HaCaT) cell line was purchased from Cell Lines Service (Eppelheim, Germany). The cells were cultured in Dulbecco’s Modified Eagle’s Medium containing 10% fetal bovine serum, 10 mM HEPES, and 1% of penicillin (1 × 10^4^ units/mL)-streptomycin (1 × 10^4^ µg/mL). Cells were maintained in a humidified incubator containing 5% CO_2_ at 37 °C throughout the experiment.

### 2.8. Cell Viability Assay

The cytotoxicity of HCFE and DPM was assessed using the MTT assay [25]. The HaCaT cells were seeded in a 96-well plate (1 × 10^4^ cells/well) and incubated in the incubator that maintained a 5% CO_2_ atmosphere at 37 °C. The seeded cells were exposed to HCFE or DPM of 6.25, 12.5, 25, 50, 100, and 200 μg/mL for 24 h. Then, MTT (50 μg/mL) was added to each well and incubated for 4 h. After removing the cell supernatant, the generated crystals were dissolved in DMSO, and the absorbance was measured at 540 nm using a SpectraMAX 190 microplate reader (Molecular Devices, CA, USA).

### 2.9. Measurement of Reactive Oxygen Species (ROS) Production

The intracellular ROS levels were measured using 2′,7′-dichlorodihydrofluorescein diacetate (DCF-DA) [26]. HaCaT cells were seeded in 48-well plates (3 × 10^4^ cells/well) and incubated overnight. The cells were pre-treated with HCFE (25, 50, and 100 μg/mL) for 30 min. Subsequently, the cells were stimulated with DPM (100 μg/mL) for 30 min and stained with DCF-DA at 20 μM for 1 h at 37 °C in the dark. After rinsing three times with phosphate-buffered saline, ROS levels were analyzed using the EVOS FLoid Color Imaging Systems (Thermo Fisher Scientific, Waltham, MA, USA). Cell fluorescence intensity was measured using ImageJ software (National Institutes of Health, Bethesda, MD, USA, v1.54).

### 2.10. RNA Extraction and Real-Time Quantitative Polymerase Chain Reaction (PCR)

Total RNA from HaCaT cells was isolated using RiboEX reagent (GeneAll Biotechnology, Seoul, Republic of Korea), and quantified using a BioSpectrometer (Eppendorf, Hamburg, Germany). Next, 2 μg of isolated RNA was used for cDNA synthesis. A HelixCript Easy cDNA Synthesis Kit (NanoHelix, Daejeon, Republic of Korea) was used to synthesize cDNA according to the manufacturer’s protocol. The cDNA was synthesized using a SimpliAmp Thermal Cycler (Applied Biosystems, Foster City, CA, USA). The synthesized cDNA was amplified using RealHelix Premier qPCR kit (NanoHelix, Daejeon, Republic of Korea), and this process was carried out by the StepOnePlus Real-Time PCR System (Applied Biosystems, Foster City, CA, USA). The primers used for amplification are shown in Table 1, and all the levels of mRNA were normalized to GAPDH.

### 2.11. Western Blotting

The total proteins were extracted using a radioimmunoprecipitation assay buffer containing Halt^TM^ protease inhibitor cocktail (Thermo Fisher Scientific, Cleveland, OH, USA). Extracted proteins were quantified using the Bradford method. SDS-polyacrylamide gel electrophoresis was performed for protein separation, and the separated proteins were transferred onto polyvinylidene fluoride membranes. Membranes were blocked with 5% skim milk (*w*/*v*) dissolved in phosphate-buffered saline containing 0.1% Tween 20 (PBST) for 1 h at room temperature. After blocking, the membranes were incubated overnight with diluted primary antibodies at 4 °C. All the antibodies were diluted to the manufacturer’s recommended concentration. The primary antibodies were removed, and the membranes were washed thrice with PBST. Subsequently, the membranes were incubated with horseradish peroxidase-conjugated secondary antibodies at room temperature for 1 h, and the proteins were detected with the enhanced chemiluminescence reagent (Advansta, San Jose, CA, USA) using the ChemiDoc imaging system (Bio-Rad Laboratories, Hercules, CA, USA).

### 2.12. Statistical Analysis

All data are expressed as mean ± standard deviation (SD), and all analyses were performed using GraphPad Prism version 8.0 (GraphPad Software, San Diego, CA, USA). Comparisons were carried out using a one-way analysis of variance (ANOVA), followed by Tukey’s multiple comparison test to evaluate statistically significant differences between the control group and various groups. All experiments were repeated at least three times, and the values were considered statistically significant at *p* < 0.05; the *p*-values were as follows: * *p* < 0.05, ** *p* < 0.01, *** *p* < 0.001.

## 3. Results

### 3.1. Ultra Performance Liquid Chromatography Analysis of Hibiscus cannabinus L. Flower Ethanol Extract (HCFE)

Myricetin-3-*O*-glucoside, a bioactive compound present in HCFE, was detected using ultra-performance liquid chromatography (Figure 1). The amount of myricetin-3-*O*-glucoside was quantified using the constructed standard curve, and the amount of myricetin-3-*O*-glucoside contained per 1 g of dry-weight HCFE was 2.43 ± 0.06 mg (Table 2).

### 3.2. Total Anthocyanin Content of HCFE

The total anthocyanin content in HCFE was measured using the pH differential method described above and calculated as cyanidin-3-*O*-glucoside equivalents. The total anthocyanin content per 100 mg of dry-weight HCFE was 1.53 ± 0.031 mg after three replicates (Table 2).

### 3.3. Antioxidant Effect of HCFE in DPM-Stimulated HaCaT Cells

For cell experiments, the cytotoxicity of DPM and HCFE to HaCaT cells was measured by MTT assay. DPM at 6.25, 12.5, 25, 50, and 100 μg/mL did not show cytotoxicity, and HCFE did not show cytotoxicity at all concentrations (6.25, 12.5, 25, 50, 100, and 200 μg/mL) up to 200 μg/mL (Figure 2A,B). The antioxidant effects of HCFE were evaluated using a DPPH radical scavenging assay. The antioxidant effect of 100 μg/mL of HCFE was almost similar to that of ascorbic acid, and the antioxidant effect of 12.5 μg/mL of HCFE was about 20% compared to that of ascorbic acid (Figure 2C). Subsequently, intracellular ROS levels were measured. Oxidative stress was induced in HaCaT cells by DPM treatment, and intracellular ROS levels were reduced in a dose-dependent manner by HCFE treatment (Figure 2D,E).

### 3.4. Anti-Inflammatory and Anti-Atopic Dermatitis Effects on HCFE Through Reduced Gene Expression in DPM-Stimulated HaCaT Cells

To confirm the anti-inflammatory and anti-atopic dermatitis effects of HCFE, the gene expression levels of inflammatory mediators were evaluated. The expressions of IL-1β, IL-6, IL-8, and MCP-1 decreased in a dose-dependent manner in DPM-stimulated HaCaT cells (Figure 3A–D). In addition, the gene expression levels of IL-4, IL-13, CXCL10, CCL5, CCL17, and CCL22, which are related to atopic dermatitis, were significantly decreased by HCFE in DPM-stimulated HaCaT cells (Figure 3E–I).

### 3.5. Inhibitory Effect of HCFE on MAPK Phosphorylation in DPM-Stimulated HaCaT Cells

To further investigate the anti-inflammatory mechanisms of HCFE, the phosphorylation levels of ERK, JNK, and p38 were analyzed. The levels of p-ERK, p-JNK, and p-p38, which are the phosphorylated forms of ERK, JNK, and p38, respectively, were significantly increased by DPM but were decreased by HCFE in HaCaT cells (Figure 4A). In particular, p38 activation decreased to the greatest extent compared to ERK and JNK (Figure 4B–D).

### 3.6. Inhibitory Effect of HCFE on NF-κB Translocation to the Nucleus in DPM-Stimulated HaCaT Cells

The translocation of p65 from the cytoplasm to the nucleus was analyzed by Western blotting. Nuclear p65 and p-p65 levels, which were increased by DPM, were markedly reduced in a dose-dependent manner after the treatment of HaCaT cells with HCFE (Figure 5A). When HaCaT cells were treated with 25 μg/mL of HCFE, there was no significant difference in the level of intranuclear p-p65 compared to DPM-only stimulation. However, when HaCaT cells were treated with HCFE 50 and 100 μg/mL, they were reduced by 37% and 65%, respectively, compared to DPM-only treatment (Figure 5B). The level of intranuclear p65 was also reduced by 59% when HaCaT cells were treated with 100 μg/mL (Figure 5C). Additionally, cytoplasmic p-IκBα was decreased by HCFE in DPM-stimulated HaCaT cells (Figure 5A). A total of 100 μg/mL of HCFE reduced the level of cytoplasmic p-IκBα by 66% in DPM-stimulated HaCaT cells (Figure 5D). However, HCFE did not affect cytoplasmic IκBα (Figure 5E).

### 3.7. NLRP3 Inflammasome Down-Regulation by HCFE in DPM-Stimulated HaCaT Cells

IL-1β, a pro-inflammatory cytokine, is associated with the activation of the NLRP3 inflammasome [27]. To determine whether HCFE affects NLRP3 inflammasome activation, NLRP3, caspase-1, and IL-1β levels were evaluated in HaCaT cells after DPM stimulation. NLRP3 levels, which increased after DPM stimulation, decreased in a dose-dependent manner in the HCFE-treated group (Figure 6A,B). In the HCFE 25 μg/mL-treated group, caspase-1 showed no significant difference in response to DPM stimulation; however, caspase-1 was decreased in the HCFE 50 and the 100 μg/mL-treated group. In particular, in the 100 μg/mL-treated group, caspase-1 and the mature form of IL-1β (cleaved IL-1β) were one-quarter of those in the group stimulated only with DPM (Figure 6A,C,D).

### 3.8. Inhibitory Effect of HCFE on JAK1/STAT6 Pathway in DPM-Stimulated HaCaT Cells

IL-4 and IL-13 are closely related to atopic dermatitis, and they activate the intracellular JAK1/STAT6 pathway [28]. The effect of HCFE on the intracellular JAK1/STAT6 pathway was analyzed using Western blotting. The levels of phosphorylated JAK1 and STAT6 in DPM-stimulated HaCaT cells were significantly reduced by HCFE treatment (Figure 7A). Phosphorylation of JAK1 and STAT6 was reduced by more than 50% compared to the control (DPM-only treatment) in the HCFE 50 and 100 μg/mL-treated group (Figure 7B,C).

### 3.9. Up-Regulation of Skin Moisturizing Factors by HCFE in DPM-Stimulated HaCaT Cells

The expression of filaggrin and involucrin was measured to investigate the effect of HCFE on skin moisture. In the DPM-stimulated group, filaggrin and involucrin levels were lower than those in the normal group, but they were significantly higher in the HCFE-treated group (Figure 7D). When treated with 100 μg/mL HCFE, the filaggrin level decreased by DPM stimulation was recovered to the level of the normal group, and the involucrin level, which was decreased by DPM stimulation, was increased above the normal group (Figure 7E,F).

## 4. Discussion

Atopic dermatitis is a common disease in children and is difficult to prevent or cure. This study aimed to demonstrate the anti-inflammatory and anti-atopic dermatitis effects of *H. cannabinus* in a DPM-stimulated in vitro model. To investigate the atopic dermatitis-improving effect of *H. cannabinus*, we measured intracellular ROS levels and the expression levels of pro-inflammatory cytokines and atopic dermatitis-related chemokines in DPM-stimulated HaCaT cells and analyzed the cell signaling mechanisms related to inflammation in the skin.

Atopic dermatitis is caused by complex interactions involving genetic factors, environmental factors, and immunological abnormalities [29]. The prominent feature of atopic dermatitis is severe itching; the vicious cycle of itching–scratching–itching causes not only skin wounds but also problems such as insomnia, learning disabilities, and decreased external activities [30,31,32]. At present, moisturizers, topical steroids, and biological agents are used to treat atopic dermatitis, but there is no perfect cure for it [33,34].

Recently, increased air pollution due to industrial development has emerged as one of the causes of various diseases such as cardiovascular, respiratory, and skin diseases [35,36,37]. Small particulate matter that can penetrate cells poses a great risk to the human body, and this has been reported to be significantly related to an increase in the prevalence of atopic dermatitis [38].

*H. cannabinus* has been primarily used in the textile industry and is called one of the world’s three major textile crops. Mainly as a fiber crop, its stem is the most commonly used; however, the use of other parts, such as roots, seeds, and flowers, has been limited until now. Previous studies have reported the effects of *H. cannabinus* seeds and leaves [39]. *H. cannabinus* seed extract has been reported to have anticancer, antibacterial, antithrombotic, and antioxidant effects [40,41,42]. In addition, *H. cannabinus* leaf extract has antioxidant, anti-inflammatory, antibacterial, antifungal, and skin-whitening effects [43,44]. *H. cannabinus* stem extract has antioxidant effects, and *H. cannabinus* flower extract has antioxidant, antibacterial, and anticancer effects; however, there is no research on *H. cannabinus* flower and its effect on the skin [20,45]. The effects of *H. cannabinus* are mainly mediated by components such as phenols, flavonoids, and anthocyanins, whereas *H. cannabinus* flowers contain anthocyanins, caffeic acid, and myricetin glycosides [20]. Moreover, the antioxidant and anti-inflammatory effects of myricetin have been well known [46,47]. In our study, the HCFE contained 2.43 ± 0.06 mg/g of myricetin, and the total anthocyanin content per 100 mg of dry-weight HCFE was 1.53 ± 0.031 mg (Figure 1 and Table 2).

Oxidative stress plays an important role in the induction and development of inflammation. Chronic inflammation can cause cardiovascular diseases, neurodegenerative diseases, and cancer [48]. In addition, atopic dermatitis is a chronic inflammatory disease of the skin, and inflammatory mediators are the main targets of treatment [49]. The effects of oxidative stress on atopic dermatitis are well known. Oxidative stress caused by ROS and nitrogen species directly damages the cell membranes, DNA, and organelles. In addition, it generates pro-inflammatory cytokines, that induce skin inflammation and damage the skin barrier [50]. When HaCaT cells were exposed to DPM, the amount of intracellular ROS significantly increased. However, the amount of intracellular ROS was greatly reduced in HaCaT cells treated with HCFE, and the intracellular ROS level in 100 μg/mL of HCFE-treated HaCaT cells was almost similar to that of the HCFE-untreated control group (Figure 2). These results demonstrate that HCFE contains antioxidant substances, such as anthocyanins and myricetin, which reduce intracellular oxidative stress in HaCaT cells exposed to DPM.

Inflammation is the body’s primary defense mechanism against tissue damage, localizing the damage and restoring it to normal; however, chronic inflammation can cause serious diseases such as cardiovascular, neurodegenerative, and autoimmune diseases [51,52]. Atopic dermatitis is caused by chronic inflammation of the skin tissue, and the onset of atopic dermatitis is closely related to various inflammatory mediators such as IL-4, IL-13, CCL5, CCL17, and CCL22 [53]. According to previous studies, Particulate matter that penetrates cells increases the production of pro-inflammatory cytokines, causing skin inflammation [54]. The increased expression of IL-1β, IL-6, IL-8, and MCP-1 by DPM exposure was observed in HaCaT cells. In addition, the expression of IL-4, IL-13, CXCL10, CCL5, CCL17, and CCL22, which are atopic dermatitis mediators, increased 2- or 3-fold in HaCaT cells treated with DPM compared with that in the control group. However, the expression of increased inflammatory mediators was significantly decreased in a dose-dependent manner by HCFE treatment, and the expression levels of all inflammatory mediators except CCL17 were reduced to normal levels in the HCFE 100 μg/mL-treated group (Figure 3). These data demonstrate that DPM induces skin inflammation, and HCFE exhibits an anti-inflammatory effect that inhibits the expression of cytokines and chemokines associated with inflammation and atopic dermatitis.

The activation of upstream signaling pathways is required for the expression of intracellular cytokine genes. MAPKs (ERK, JNK, and p38) and NF-kB are representative signaling pathways involved in the expression and secretion of pro-inflammatory cytokines [55,56]. In previous studies, ERK and p38 were shown to induce IL-6 and IL-8 expression [57,58]. In addition, p65, which is translocated into the nucleus and phosphorylated, regulates the expression of inflammatory cytokines and chemokines such as IL-1, IL-2, IL-6, and IL-8 [55]. In our study, a decrease in MAPK phosphorylation by HCFE was observed in DPM-stimulated HaCaT cells, and p38 phosphorylation was significantly reduced in the group treated with high concentrations of HCFE. Similarly, nuclear p65 and phosphorylated p65 were also reduced after HCFE treatment, and a decrease in cytoplasmic phosphorylated IκBα was also observed (Figure 4 and Figure 5). These results show that HCFE suppresses gene expression of inflammatory cytokines and chemokines through down-regulation of MPAKs and NF-κB signaling.

IL-1β, an inflammatory cytokine, is produced as a precursor within cells and is processed by the NLRP3 inflammasome to form the final IL-1β, which is secreted out of the cells [59]. Particulate matter infiltrates cells, increases intracellular ROS, and activates the NF-κB pathway to induce NLRP3 inflammasome expression [60]. Figure 6 shows the inhibitory effects of HCFE on the NLRP3 inflammasome. HCFE significantly reduced the levels of NLRP3 and caspase-1, and components of the NLRP3 inflammasome, that were increased by DPM exposure in HaCaT cells. These results imply that HCFE controls the secretion of IL-1 by suppressing NLRP3 inflammasome activity.

An increase in Th2 cytokines, including IL-4 and IL-13, is an important feature in atopic dermatitis lesions [61]. IL-4 and IL-13 activate the JAK1/STAT6 pathway after interacting with their receptors [28]. Another important feature of atopic dermatitis is the collapse of the skin barrier. Innate or acquired mutations in filaggrin or deficiencies in filaggrin and involucrin cause transepidermal water loss [11,62]. JAK1 and STAT6 are phosphorylated upon activation, and HCFE inhibits the phosphorylation of JAK1 and STAT6 in a dose-dependent manner in DPM-stimulated HaCaT cells. Furthermore, HCFE restored the loss of filaggrin caused by DPM exposure and enhanced involucrin expression above that in the control group (Figure 7). These data indicate that HCFE inhibits the JAK1/STAT6 pathway and induces reconstruction of the collapsed skin barrier by DPM exposure.

Our findings show that HCFE reduces skin inflammation and protects skin by regulating inflammatory cytokines/chemokines through the inhibition of MPAK, NF-kB, and inflammasome activation, blocking Th2 cytokine signaling by down-regulating the JAK1/STAT6 pathway and protecting the skin barrier by inhibiting the loss of filaggrin and involucrin (Figure 8). Based on these results, we suggest that *H. cannabinus* flower extract could be used as a therapeutic agent for skin inflammation and as a natural functional food for alleviating and treating skin inflammation symptoms.

## 5. Conclusions

We have demonstrated the anti-inflammatory and anti-atopic dermatitis effects of *H. cannabinus* flower extract (HCFE) through an intracellular mechanism analysis in skin inflammation in vitro models. Down-regulation of MAPKs, NF-kB, inflammasome, and JAK1/STAT6 pathways was observed in cells treated with HCFE, which induced the suppression of cytokine/chemokine expression and secretion. In contrast, the up-regulation of filaggrin and involucrin was observed in cells treated with HCFE. Our findings suggest that HCFE could play a major role in the improvement of atopic dermatitis symptoms caused by chronic inflammation, skin barrier disruption, and skin moisture loss. Additionally, we showed that HCFE could help improve the treatment of skin inflammatory diseases caused by changes in the atmospheric environment.

## Figures and Tables

**Figure 1 nutrients-16-03805-f001:**
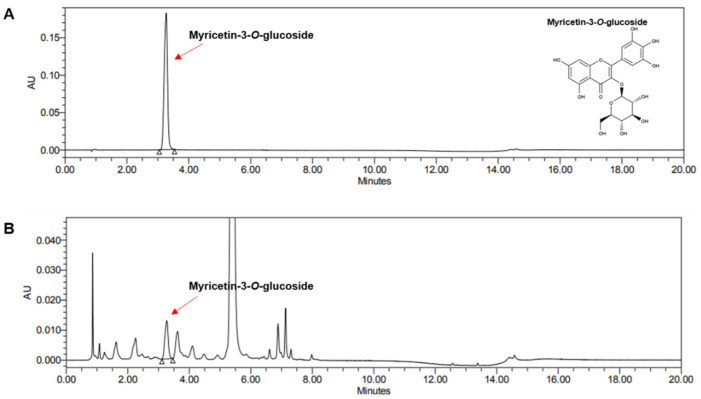
Ultra Performance Liquid Chromatography (UPLC) analysis of HCFE. The UPLC chromatograms of (**A**) myricetin-3-*O*-glycoside standard and (**B**) Extract of *Hibiscus cannabinus* L. flower.

**Figure 2 nutrients-16-03805-f002:**
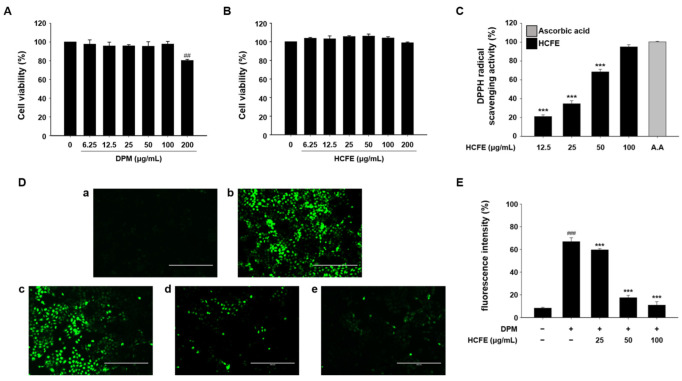
Antioxidant effect of *H. cannabinus* flower ethanol extract (HCFE) and cytotoxicity of Diesel Particulate Matter (DPM) and HCFE in HaCaT cells. HaCaT cells were treated with different concentrations (6.25, 12.5, 25, 50, 100, and 200 μg/mL) of DPM and HCFE for 24 h. The cytotoxicity of DPM and HCFE was measured by MTT assay. The graphs refer to the cytotoxicity of DPM (**A**) and HCFE (**B**) in HaCaT cells. Solvent control is 0 μg/mL. (**C**) DPPH radical scavenging activity of HCFE (12.5, 25, 50, and 100 μg/mL). Ascorbic acid (A.A) 20 μg/mL was the control. (**D**) HaCaT cells were pre-treated with HCFE 25, 50, and 100 μg/mL for 30 min, and stimulated with DPM 100 μg/mL for 30 min. The level of ROS was assessed using a fluorescence microscope (×200 magnification, scale bar = 400 μm), (**a**) control (unexposed cells), (**b**) DPM-exposed cells, (**c**) DPM and HCFE 25 μg/mL co-exposed cells, (**d**) DPM and HCFE 50 μg/mL co-exposed cells, and (**e**) DPM and HCFE 100 μg/mL co-exposed cells. (**E**) The fluorescence intensity was expressed as the percentage of cells with bright fluorescence. The data were expressed as means ± SD (*n* = 3). ^##^
*p* < 0.01 and ^###^
*p* < 0.001 vs. control group; *** *p* < 0.001 vs. DPM-stimulated group.

**Figure 3 nutrients-16-03805-f003:**
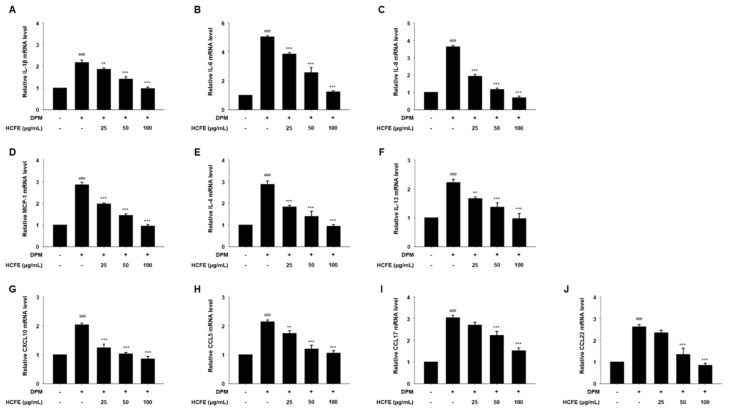
Inhibitory effect of *H. cannabinus* flower ethanol extract (HCFE) on gene expression of inflammatory mediators in HaCaT cells. HaCaT cells were pre-treated with the indicated concentration of HCFE (25, 50, and 100 μg/mL) for 30 min, and stimulated with Diesel Particulate Matter (DPM) 100 μg/mL. The mRNA levels of (**A**) IL-1β, (**B**) IL-6, (**C**) IL-8, (**D**) MCP-1, (**E**) IL-4, (**F**) IL-13, (**G**) CXCL10, (**H**) CCL5, (**I**) CCL17, and (**J**) CCL22 were measured using quantitative real-time PCR and normalized to GAPDH. The data were represented as means ± SD (*n* = 3). ^###^
*p* < 0.001 vs. control group; ** *p* < 0.01 and *** *p* < 0.001 vs. DPM-stimulated group.

**Figure 4 nutrients-16-03805-f004:**
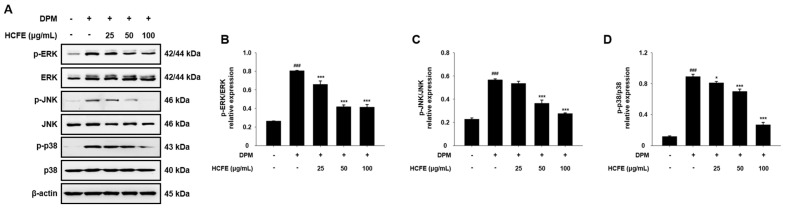
MAPKs (ERK, JNK, p38) inhibitory effect of *H. cannabinus* flower ethanol extract (HCFE) on HaCaT cells. HaCaT cells were pre-treated with the indicated concentration of HCFE (25, 50, and 100 μg/mL) for 30 min, and stimulated with Diesel Particulate Matter (DPM) 100 μg/mL. (**A**) The protein levels of ERK, JNK, and p38 were evaluated by Western blotting. The graphs were expressed as the ratio of (**B**) p-ERK/ERK, (**C**) p-JNK/JNK, and (**D**) p-p38/p38. The data were represented as means ± SD (*n* = 3). ^###^
*p* < 0.001 vs. control group; * *p* < 0.05 and *** *p* < 0.001 vs. DPM-stimulated group.

**Figure 5 nutrients-16-03805-f005:**
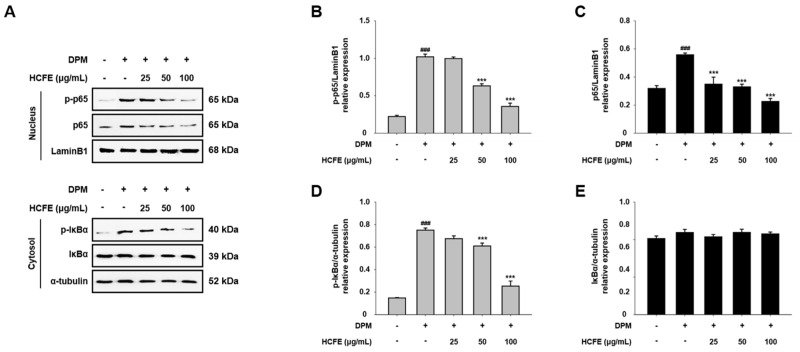
Inhibition of p65 nuclear translocation by *H. cannabinus* flower ethanol extract (HCFE) in HaCaT cells. HaCaT cells were pre-treated with different concentrations of HCFE (25, 50, and 100 μg/mL) for 30 min and then stimulated with Diesel Particulate Matter (DPM) 100 μg/mL. Cell fractionation was performed after the whole protein harvest using the Nuclear and Cytoplasmic Extraction Kit. (**A**) The protein levels of p-p65 and p65 in the nucleus and IκBα and p-IκBα in the cytoplasm were assessed by Western blotting. Lamin B1 and α-tubulin were used as loading controls for the nucleus and cytoplasm. Quantification graphs (relative protein expression) were expressed as the density ratio of (**B**) p-p65, (**C**) p65, (**D**) p-IκBα, and (**E**) IκBα to loading controls. The data were represented as means ± SD (*n* = 3). ^###^
*p* < 0.001 vs. control group; *** *p* < 0.001 vs. DPM-stimulated group.

**Figure 6 nutrients-16-03805-f006:**
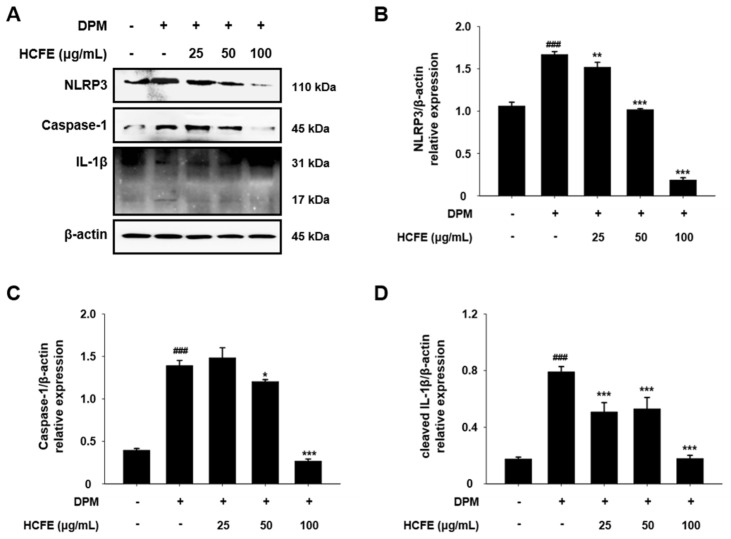
Inhibition of NLRP3 inflammasome by *H. cannabinus* flower ethanol extract (HCFE) in HaCaT cells. HaCaT cells were pre-treated with different concentrations of HCFE (25, 50, and 100 μg/mL) for 30 min and then stimulated with Diesel Particulate Matter (DPM) 100 μg/mL. (**A**) The protein levels of NLRP3, caspase-1, and IL-1β were assessed by Western blotting. Quantification graphs (relative protein expression) were expressed as the density ratio of (**B**) NLRP3, (**C**) caspase-1, and (**D**) cleaved IL-1β (17 kDa) to β-actin. The data were represented as the means ± SD (*n* = 3). ^###^
*p* < 0.001 vs. control group; * *p* < 0.05, ** *p* < 0.01 and *** *p* < 0.001 vs. DPM-stimulated group.

**Figure 7 nutrients-16-03805-f007:**
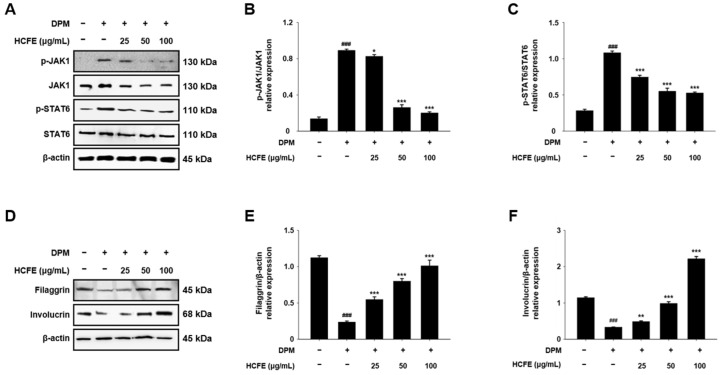
Anti-atopic dermatitis effects of *H. cannabinus* flower ethanol extract (HCFE) through regulation of JAK1/STAT6 and skin moisturizing factors. HaCaT cells were pre-treated with the indicated concentration of HCFE (25, 50, and 100 μg/mL) for 30 min and then stimulated with Diesel Particulate Matter (DPM) 100 μg/mL. The protein levels of (**A**) p-JAK1, JAK1, p-STAT6, STAT6, (**D**) filaggrin, and involucrin were assessed by Western blotting. Quantification graphs were expressed as the ratio of (**B**) p-JAK1/JAK1, (**C**) p-STAT6/STAT6, (**E**) filaggrin/β-actin, and (**F**) involucrin/β-actin. The data were represented as means ± SD (*n* = 3). ^###^
*p* < 0.001 vs. control group; * *p* < 0.05, ** *p* < 0.01 and *** *p* < 0.001 vs. DPM-stimulated group.

**Figure 8 nutrients-16-03805-f008:**
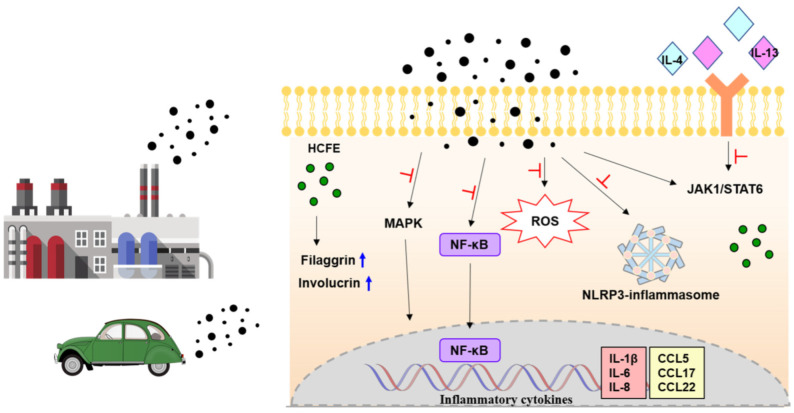
Scheme of mechanism of anti-inflammatory effects by *H. cannabinus* flower ethanol extract (HCFE) in HaCaT cells. HCFE inhibits various steps in the skin inflammation mechanism. MAPK: Mitogen-activated protein kinase; NF-κB: Nuclear factor kappa B; ROS: Reactive oxygen species; NLRP3: NLR Family Pyrin Domain Containing 3; JAK1: Janus kinase 1; STAT6: Signal transducer and activator of transcription 6.

**Table 1 nutrients-16-03805-t001:** Primer list for real-time PCR.

Gene		Primer Sequence
*IL-1β*	Forward:	5′-CTCTCTCACCTCTCCTACTCAC-3′
	Reverse:	5′-ACACTGCCTACTTCTTGCCCC-3′
*IL-4*	Forward:	5′-ACATTGTCACTGCAAATCGACACC-3′
	Reverse:	5′-TGTCTGTTACGGTCAACTCGGTGC-3′
*IL-6*	Forward:	5′-CTCCAC AAGCGCCTTCGGTC -3′
	Reverse:	5′-TGTGTGGGGCGGCTACATCT-3′
*IL-8*	Forward:	5′-ACCGGAGCACTCCATAAGGCA-3′
	Reverse:	5′-AGGCTGCCAAGAGAGCCACG-3′
*IL-13*	Forward:	5′-ACCACGGTCATTGCTCTCACT-3′
	Reverse:	5′-GTCAGGTTGATGCTCCATAC-3′
*MCP-1*	Forward:	5′-TCTGTGCCTGCTGCTCATAG-3′
	Reverse:	5′-CAGATCTCCTTGGCCACAAT-3′
*CXCL10*	Forward:	5′-TTGCTGCCTTATCTTTCTGACTC-3′
	Reverse:	5′-ATGGCCTTCGATTCTGGATT-3′
*CCL5*	Forward:	5′-CGCTGTCATCCTCATTGCTA-3′
	Reverse:	5′-GCACTTGCCACTGGTGTAGA-3′
*CCL17*	Forward:	5′-CCATTCCCCTTAGAAAGCTG-3′
	Reverse:	5′-CTCTCAAGGCTTTGCAGGTA-3′
*CCL22*	Forward:	5′-TGCCGTGATTACGTCCGTTAC-3′
	Reverse:	5′-AAGGCCACGGTCATCAGAGTAG-3′
*GAPDH*	Forward:	5′-GAAGGTGAAGGTCGGAGT-3′
	Reverse:	5′-GAAGATGGTGATGGGATTTC-3′

**Table 2 nutrients-16-03805-t002:** Total anthocyanin content and Myricetin-3-*O*-glucoside of *Hibiscus cannabinus* L. flower extract (HCFE).

Sample	Absorbance				A′	Content (mg/100 mg)	Myricetin-3-O-Glucoside (mg/g)
	pH 1.0		pH 4.5				
	λ 520 nm	λ 700 nm	λ 520 nm	λ 700 nm			
HCFE	0.250	0.019	0.072	0.020	0.178	1.490	2.43 ± 0.06
	0.258	0.020	0.072	0.020	0.186	1.553	
	0.255	0.019	0.064	0.014	0.187	1.557	
Average						1.53 ± 0.031	

A′ is absorbance = (520 nm pH 1.0 − 700 nm pH 1.0) − (520 nm pH 4.5 − 700 nm pH 4.5).

## Data Availability

The data used to support the findings of this study are included in the article.

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
