# Peer review of "Anti-Inflammatory Effect of Ethanol Extract from Hibiscus cannabinus L. Flower in Diesel Particulate Matter-Stimulated HaCaT Cells"

_nutrients, 2024, doi:10.3390/nu16223805_

Round 1

Reviewer 1 Report

Comments and Suggestions for Authors

The in vitro study by Han et al. examined the anti-inflammatory actions of Hibiscus extract in diesel particle matter-stimulated HaCat Cells.

The report is interesting, but there are many minor and some major issues that need to be addressed.

Minor:

1.Line 34 need reference for main cause of air pollution

2.Line 43 recurrence is likely

3.Line 55 need reference for filaggrin

4.Line 62, Therefore, better treatments that have……are needed

5. Line 64 Herbal medicine has been used…..

6. Line 67 awkward sentence “nature-friendly trend grows”- please rewrite

7. Line 73 need reference for fiber crop worldwide

8. Line 80 Overstatement the authors did not investigate atopic dermatitis, please re-write

9. Line 194 city is missing

10. Please describe/define size of DPM in section 2.2

11. Section 2.7 use of 10 % fetal bovine serum contains steroids (esp. estradiol) how did the authors control for this protective hormone in the in vitro studies

Major

1.  Line 81 Rewrite with a clear hypothesis for this in vitro study

2. Figure 2 D labels are too small to read, enlarge the font size for the labels

3. Take out bold font terms throughout the text….

4. The authors need to justify how a very high (in vitro) dose of 100 microgram/ml of HCFE can be used to treat dermal conditions and need to address the following questions:

a. what is the stability of HCFE in a standard dermal formulation

b. what is the penetration of HCFE into human skin

c. what are the examples of HCFE in dermal studies performed by previous investigators

d. what is the potential influence of all the other active compounds in the HCFE extract  that are not accounted for in the present study that may contribute to positive actions for anti-inflammatory effects

5. Throughout the discussion (section) the authors make several/many overstatements, jumping from the in vitro results to treating atopic dermatitis and how HCFE can be used to treat atopic dermatitis, which is inappropriate…  in other words tone down the claims from the obtained in vitro results to dermal disease treatments.

6. Line 458 Awkward sentence, please re-write,…….the pathogenic mechanism studied in the present in vitro study may be involved in inflammatory conditions like atopic dermatitis…

Comments on the Quality of English Language

This report need many minor revisions for English gramma and sentence/phase text issues

Author Response

Thank you for your feedback on our manuscript. 

Reviewer 2 Report

Comments and Suggestions for Authors

Here, an extract from Hibiscus cannabinus L. flower was shown to have an anti-inflammatory effect on HaCaT cells exposed to diesel particle matter. Overall, the scientific quality of the manuscript is good. The results are relevant to the field and provide an appropriate background for future studies on the topic. I would like to suggest some points for the improvement of the present work.

1. In the present study, the authors used the flowers of Hibiscus cannabinus L., but there are several studies with other plant parts as well. In the Discussion (Lines 366-377), this aspect could be explored in more detail. Please check the review (doi: 10.1016/j.foodchem.2020.128582) and other related studies to get some insights into this aspect.

2. The concentrations of the extract and DPM used in the assays should be provided in the abstract. Moreover, justify/reference the use of the concentrations in the Methods section.

3. In the figure legends, please include the full description of DPM and other abbreviations.

4. I believe that the authors should have also investigated the IL-1β levels via western blotting in Figure 6. This addition would greatly help in the discussion of the results, especially when the authors state that “These results imply that HCFE controls the secretion of IL-1 by suppressing NLRP3-inflammasome activity.” (Lines 429-430).

5. In section 2.1, provide the specific dilutions used for each antibody.

6. Some of the methods could have more details, especially Section 2.3.

7. Include the kDa of the proteins in the western blot representative bands.

8. In Figures 2A and 2B the control group of cells seems to not have SD, please check. The same applies to graphs in Figure 3.

9. A representative figure with proposed mechanisms of action of the extract is suggested to increase the searchability of the manuscript.

10. Revise abbreviations used throughout the document. Also, scientific names should be provided in full at their first mention and then abbreviated throughout. For example, “Hibiscus cannabinus” at first mention and “H. cannabinus” throughout.

Author Response

(The authors gave the same response as above.)

Reviewer 3 Report

Comments and Suggestions for Authors

The manuscript described the anti-inflammatory effect of ethanol extract of Hibiscus cannabinus L. flower (HCFE) in Diesel Particle Matter (DPM)-induced HaCaT cells, and also the contents of bioactive factors and the potential mechanisms. Numerous studies have revealed the chemical constituents and bioactivities of different parts from H. cannabinus, the authors studied the effects of HCFE on DPM-induced HaCaT cells, which related with atopic dermatitis. Thus, the results are some of interest and would improve the application of HCFE on skin disease. Before the paper acceptance, there are some concerns as following.

1.     What’s the particle size of H. cannabinus L. flower in extract experiment 2.3? And what’s the extract rate?

2.     The wavelength of 356 nm used in HPLC analysis should be stated in Figure 1. And why choose this wavelength?

3.     What’s the vehicle for HCFE in HaCaT cells? Did the authors check the effects of vehicle solvent in HaCaT cells?

4.     For the bioactivity of HCFE, why the authors check ROS and DPPH activity instead of the inflammatory factor, such as interleukins?

Author Response

(The authors gave the same response as above.)

Round 2

Reviewer 1 Report

Comments and Suggestions for Authors

The authors have addressed all the reviewers comments, etc.